# The Effects of Short-Chain Fatty Acids in Gut Immune and Oxidative Responses of European Sea Bass (*Dicentrarchus labrax*): An Ex Vivo Approach

**DOI:** 10.3390/ani14091360

**Published:** 2024-04-30

**Authors:** Filipa Fontinha, Nicole Martins, Gabriel Campos, Helena Peres, Aires Oliva-Teles

**Affiliations:** 1Departamento de Biologia, Faculdade de Ciências, Universidade do Porto, Rua do Campo Alegre s/n, Edifício FC4, 4169-007 Porto, Portugal; filipafontinha@hotmail.com (F.F.); martinspnicole@gmail.com (N.M.); gabrielcrcampos00@gmail.com (G.C.); pereshelena@fc.up.pt (H.P.); 2Centro Interdisciplinar de Investigação Marinha e Ambiental (CIIMAR), Universidade do Porto, Terminal de Cruzeiros do Porto de Leixões, Avenida General Norton de Matos, s/n 289, 4450-208 Matosinhos, Portugal

**Keywords:** ex vivo, functional ingredients, short-chain fatty acids, immune status

## Abstract

**Simple Summary:**

SCFAs are considered functional ingredients as they promote the growth and health of fish. This work aimed to assess the effect of short-chain fatty acids (SCFAs) on the immune status of the European sea bass gut through the ex vivo use of intestinal explants. The ex vivo technique significantly reduces the number of animals to be used, according to the guidelines of the 3Rs. The present results are important to understand how European sea bass respond to a bacterial infection after the administration of SCFAs and for the evaluation of functional ingredients used in aquaculture.

**Abstract:**

This study aimed to evaluate the intestinal interactions between three short-chain fatty acids (SCFA), namely, acetate, propionate, and butyrate, and pathogenic bacteria (*Vibrio anguillarum*) in intestinal explants of European sea bass (*Dicentrarchus labrax*) juveniles. The anterior intestine of 12 fish with an average weight of 100 g (killed by excess anesthesia with 2-phenoxyethanol) were sampled and placed in 24-well plates. The experimental treatments consisted of a control medium and a control plus 1 mM or 10 mM of sodium acetate (SA), sodium butyrate (SB), and sodium propionate (SP). After 2 h of incubation, the explants were challenged with *Vibrio anguillarum* at 1 × 10^7^ CFU/mL for 2 h. After the bacterial challenge, and regardless of the SCFA treatment, the oxidative stress-related genus catalase (*cat*) and superoxide dismutase (*sod*) were down-regulated and glutathione peroxidase (*gpx*) was up-regulated. Furthermore, the immune-related genes, i.e., the tumor necrosis factor (*TNF-α*), interleukin 8 (*IL-8*), transforming growth factor (*TGF-β*), and nuclear factor (*NF-Kβ*) were also up-regulated, and interleukin 10 (*IL-10*) was down-regulated. During the pre-challenge, sodium propionate and sodium butyrate seemed to bind the G-protein coupled receptor (*grp40L*), increasing its expression. During the challenge, citrate synthase (*cs*) was down-regulated, indicating that the SCFAs were used as an energy source to increase the immune and oxidative responses. Overall, our results suggest that sodium propionate and sodium butyrate may boost European sea bass immune response at the intestine level.

## 1. Introduction

The rapid expansion of aquaculture has been pivotal in fulfilling the human population’s demand for high-quality protein and food security. However, this increase in aquaculture has been achieved by increasing the intensity of rearing systems, and this can induce stress and compromise fish health and immunity, increasing mortality and decreasing productivity [1].

Antibiotics are often used in aquaculture to mitigate health issues imposed by culture intensification. Nevertheless, the indiscriminate use of several antibiotics may lead to bacterial resistance, leading to a reduction in their effectiveness Efforts are being made to find alternatives to antibiotics and improve fish health and immunological status [2]. One of these alternatives is the use of functional ingredients such as short-chain fatty acids (SCFAs).

SCFAs are molecules with less than seven carbon atoms and have slightly acidic characteristics. The most abundant SCFAs are acetic, propionic, and butyric acid, as well as their salts [3]. They have long been utilized in feeds as preservatives and antimicrobial agents and are generally regarded as safe for consumption by both humans and animals, being considered functional additives [4]. In animals, SCFAs are produced by the anaerobic fermentation of non-digestible carbohydrates in the intestine [5,6].

The gastrointestinal tract is a versatile organ with several biological functions. Among these functions, physical and immunological defense against pathogenic microorganisms is one of the most important [7]. The intestinal epithelial cells are essential for mucosal immunity and are involved in the production of antimicrobial peptides, chemokines, cytokines, and immunoglobulins that help fish defend against infections [8,9].

The mode of action of SCFAs in the intestinal tract comprises different actions, such as lowering the lumen pH, enhancing the digestive enzyme activity, and creating an unfavorable environment for pathogens, thus inhibiting the growth of Gram-negative bacteria [4]. Thus, the application of SCFAs as functional ingredients in fish feed may reduce their mortality when fish are exposed to pathogenic bacteria [10].

For instance, Xun et al. (2022) [11] showed that, in golden pompano (*Trachinotus ovatus*), supplementation with sodium acetate (0, 250, 500, 1000, 2000, and 4000 mg/kg) improved the oxidative status and decreased the gene expression of some immune-related genes like *TNF-α* and *NF-kβ*. In Nile tilapia (*Oreochromis niloticus*), the addition of (100 and 200 mmol/L) sodium acetate to their diet was shown to decrease intestinal inflammation [12]. Also, in European sea bass (*Dicentrarchus labrax*) [13] and the common carp (*Cyprinus carpio* L.) [14], diets supplemented with sodium propionate (0.1, 0.2, 0.3%) and (0, 0.5, 1, 2%), respectively, improved the immune system and antioxidant status of the animals.

Furthermore, juvenile largemouth bass (*Micropterus salmonids*) fed with sodium butyrate (0, 0.5, 1, and 2 g/kg) demonstrated an increased antioxidant capacity and decreased immune function [15]. On the other hand, Liu et al. (2014) [16] showed an increased immune response in common carp fed with sodium (0, 1.5, 3%).

Most of these studies showed that SCFAs improve the immune function and protect fish against oxidative stress by increasing the antioxidant ability and scavenging of free radicals [13,17,18,19]. These anti-inflammatory and antioxidant defenses may be activated by the binding of SCFAs to G protein-coupled receptors (GPCRs) in intestinal cells [20]. Furthermore, SCFAs are absorbed in enterocytes and used as an energy source, or transported to the liver, where they are metabolized in the carboxylic acid cycle for energy production, such as ATP production [21].

However, the lack of adequate in vitro systems to study and understand the physiological processes that occur at the intestine level of several fish species has prompted scientists to find new ways to study these interactions [22]. Thus, innovative systems, like the use of explants (non-disaggregated tissue or organ fragments that have been taken from an organism), are being tested to study the interactions between the host and different compounds or bacteria [9,23].

Under suitable conditions, ex vivo techniques allow the maintenance of whole organs or fragments in viable organized and differentiated conditions. Furthermore, with the appropriate supplementation, the cells maintain biochemical signaling with their natural neighbors [24]. Moreover, by highly reducing the number of animals required in experimentation, the ex vivo technique meets the principle aims of the 3Rs, i.e., replacement, reduction, and refinement [25]. Replacement refers to a complete substitution of the use of living animals when possible. Reduction involves reducing the number of animals used in each experiment through the optimization of the experimental design, and refinement focuses on the use of practices to minimize pain, distress, and suffering, in this case, improving the welfare of aquatic animals [26,27].

In aquaculture, this may involve developing new methods that do not rely on fish, such as the use of computer models and cell cultures or the implementation of ex vivo systems; although they do not eliminate the use of animals, they highly reduce their use and avoid inducing pain, distress, and suffering to the experimental animals [26,27].

In the present work, an intestine explant culture was used to evaluate the antioxidant and immune responses of European sea bass to a pathogenic bacterial challenge after exposure to SCFAs.

## 2. Materials and Methods

### 2.1. Bacteria Culture

The culture medium used for *Vibrio anguillarum* DSMZ 21597 was brain heart infusion (BHI) (BD, Becton, Dickinson, and Company, MD21152, Sparks, NV, USA). Vibrio was grown for 24 h at 25 °C and 140 rpm. The culture was centrifugated for 15 min at 4000× *g*. Bacterial cells were resuspended in sterile BHI to reach a final concentration of 1 × 10^7^ cfu/mL. To confirm the used concentration, a serial dilution of bacterial suspension was plated, and colony-forming units were (CFUs) counted.

### 2.2. Fish Rearing Conditions

The experiment was carried out by trained scientists (following the FELASA category C recommendation) (and approved by the ORBEA Animal Welfare Committee of CIIMAR (ORBEA; reference ORBEA_CIIMAR_27_2019) according to the European Union Directive 2010/63/EU and the Portuguese Law DL 113/2013.

European sea bass (*Dicentrarchus labrax*) were fed commercial diets (Aquasoja Sustainable Feed-Standart Orange 4 MEO4; Sorgal, Ovar, Portugal; analytical constituents: 44% crude protein, 18% crude fat 2.5% fiber, and 11.1% ash) and kept in an experimental recirculated system with cylindrical fiberglass tanks of 500 L capacity. The water temperature was maintained at 22 ± 1 °C, with a salinity of 33 ± 1‰, dissolved oxygen above 80% saturation, and a photoperiod of 12 h light and 12 h dark cycle.

### 2.3. Ex Vivo Trial

In this study, 12 European sea bass juveniles with an average weight of 100 g were used. The fish were fasted for 24 h and euthanized by excess anesthesia with 2-phenoxyethanol (1 mg/mL). The anterior intestines were sampled, sliced into pieces with an average weight of 1.3 mg, and placed in 24-well plates (2 explants per well).

The experimental treatments consisted of 1 mM and 10 mM of sodium acetate (SA) (Alfa Aesar A13184, Haverhill, MA, USA), sodium propionate (SP) (Alfa Aesar A17440, Haverhill, MA, USA), and sodium butyrate (SB) (Alfa Aesar A11079, Haverhill, MA, USA), and a control (without SCFAs) was also used. Each treatment was replicated 6 times.

First, the intestinal explants were incubated with Dulbecco’s Modified Eagle Medium (DMEM) (500U penstrep, 10% FBS, 2 mM glutamine, and 5.5 mM glucose) for 1 h at 22 °C and 100 rpm. Then, the pre-treatment medium was removed, and the explants were rinsed with PBS and incubated for 2 h at 22 °C and 100 rpm with DMEM (100U penstrep, 10% FBS, 2 mM glutamine, and 5.5 mM glucose) supplemented with each SCFA at the defined concentration (1 or 10 mM) or without SCFAs (control). At the end of this period, one explant per well was collected (pre-challenge samples) and stored in RNAlater at −80 °C until analysis. The remaining explant in each well was challenged with *Vibrio anguillarum* at 1 × 10^7^ CFU/mL for 2 h at 22 °C and 100 rpm and then collected and stored in RNAlater at −80 °C until analysis (challenge samples).

### 2.4. Viability Assay

To assess explant viability, 3-(4,5-dimethylthiazol-2-yl)-2,5-diphenyltetrazolium bromide (MTT) activity was determined according to [28] at the end of the pre-challenge and challenge. MTT is a yellow solution that is reduced to purple formazan in living cells through NADH oxidoreductase-dependent enzymes.

After incubation in 24-well plates with all treatments, as described above, the explants were rinsed twice with PBS, 2 mL MTT solution (2 mg/mL) was added to each well, and the plate was incubated at 22 °C and 100 rpm for 2 h. Then, the MTT solution was removed, the wells were rinsed twice with 1 mL of PBS, and the explants were incubated with DMSO for 1 h 20 min at 100 rpm. Then, the formed formazan was measured at 570 nm using a Synergy HT microplate reader. The amount of formazan was proportional to the viable cells, and to calculate the viability, each explant was weighted, and the viability index was calculated as indicated below.

As a control, to ensure that the MTT assay accurately distinguished between viable and non-viable tissues, 3 explants were treated with 10% dimethyl sulfoxide (DMSO) before the addition of the MTT solution to inhibit formazan production, and these explants were considered non-viable.
Viability index = (formazan absorbance (abs))/(tissue weight (mg))

### 2.5. Gene Expression

The total RNA was extracted from the anterior portion of the intestine with a TRIzol reagent (Direct-zolTM RNA Miniprep, Zymo Research, R2050, Tustin, CA, USA) and Precellys evolution apparatus (Bertin Instruments, Montigny-le-Bretonneux, France). The RNA quality was determined by 1% agarose gel. Total RNA concentration was assessed by spectrophotometry (µDrop™ plate, ThermoScientific, Waltham, MA, USA) and adjusted to 0.2 µg/8 µL H_2_O for complementary DNA synthesis using the NZY First-Strand cDNA Synthesis Kit (NZYTech, MB12502, Lisbon, Portugal).

Real-time quantitative PCR analysis (CFX Connect™ Real-Time System, Bio-Rad, Hercules, CA, USA) was used to determine the relative expression of genes in the anterior intestine. The genetic markers used in this study were two pro-inflammatory markers, namely, *TNF-α* (Tumor necrosis factor alpha) and *IL-8* (Interleukin 8); two anti-inflammatory markers, namely, *TGF-β* (Transforming growth factor beta) and *IL-10* (Interleukin 10); one apoptotic marker, namely, caspase 3; one transcriptor factor, namely, *NF-kβ* (Factor nuclear kappa B); one SCFA receptor, namely, *grp40* (G-protein coupled receptor); three oxidative stress markers, namely, *cat* (catalase), *sod* (superoxide dismutase), and *gpx* (Glutathione peroxidase); and one energy metabolism marker, namely, *cs* (citrate synthase). Two reference genes were used, namely, elongation factor 1α (*ef1α*) and *40S* ribosomal RNA.

The primers were designed In Primer3, and the quality was analyzed with the Beacon Designer Program. Primer efficiency was validated with serial two-fold dilutions of cDNA and calculated from the slope of the regression line of the quantification cycle (Ct) versus the relative concentration of cDNA. *NF-kβ* was found in the European sea bass genome database (http://seabass.mpipz.mpg.de/cgi-bin/hgGateway) (accessed on 16 September 2023. The primer sequences are listed in Table 1.

The relative gene expression values are given as the normalized mean ± standard error (SE), corresponding to the ratio between the copy numbers of the target gene and the geometric mean of the copy numbers of the reference genes.

### 2.6. Statistical Analysis

The data are expressed as the mean and the standard error of the mean (SEM) of normalized values and were statistically analyzed by a three-way ANOVA with time (pre-challenge and challenge), SCFA (sodium acetate, sodium propionate, and sodium butyrate), and SCFA concentration (1 and 10 mM) as the main factors. Before performing the ANOVA, the normality and homogeneity of variances were tested by the Shapiro–Wilk and Levene tests, respectively, and normalized when necessary. The differences were considered statistically significant when *p* ≤ 0.05. Significative differences between means were determined by Tukey’s multiple-range tests. Non-orthogonal contrasts between the control and the other treatments were performed for each sampling time. All statistical analyses were performed using the SPSS 28.0 software package for Windows (IBM^®^ SPSS^®^ Statistics, Armonk, NY, USA).

## 3. Results

### 3.1. Viability

Viability was much lower in DMSO-treated tissues than in the other treatments. Within the experimental groups, viability was much higher in the control and the SCFA tested at the highest concentration (10 mM). In these groups, viability significantly decreased after the bacterial challenge, while in the 1 mM SCFA treatments, there were no differences between pre-challenge and challenge viability (Figure 1).

### 3.2. Immune-Related Genes

The gene expression of *TNF-α* increased in all groups after the challenge, and the increase was higher at the higher SCFA concentration (Figure 2A). Compared to the control, *TNF-α* expression was higher in the 10 mM SP group.

The expression of *IL-8* also increased in all groups after the challenge, but no differences were observed between the SCFA treatments both before and after the challenge (Figure 2B). Compared to the control, *IL-8* expression was higher in the 10 mM SP and SB groups at the pre-challenge and in the 1 mM SB group after the challenge.

The expression of *TGF-β* also increased in all groups after the challenge, but no differences were observed compared to the control (Figure 2C).

The expression of *NF-kβ* also increased in all groups after the challenge, and compared to the control, it was higher in all 1 mM SCFA treatments at the pre-challenge, while after the challenge, it was only higher than the control in the 1 mM SB group (Figure 2F).

On the other hand, the expression of *IL-10* decreased after the challenge in all SCFA groups, while no differences were observed compared to the control (Figure 2D). In contrast, the expression of caspase 3 was not affected by SCFA treatment or sampling point (Figure 2E). However, it was higher at the pre-challenge in the 10 mM SP group than in the control.

### 3.3. Antioxidant-Related Genes

The expression of *gpx* increased in all groups after the challenge, except for the control and the 1 mM SP groups, where no differences were observed (Figure 3A). After the challenge, *gpx* expression in the 10 mM SP group was much higher than in the control.

The expression of *cat* and *sod* decreased in all groups after the challenge, except for the control and the 10 mM SA groups, where no differences were observed (Figure 3B,C). Compared to the control, *cat* expression after the challenge was higher in the 10 mM SA group, while sod expression was higher in the 10 mM SP group before the challenge.

### 3.4. Energy Metabolism and Free Fatty Acid Receptor-Related Genes

The expression of cs decreased in all groups after the challenge, except for the control and the 10 mM SP group (Figure 4A). Compared to the control, *cs* expression in the 10 mM SP group was higher before the challenge.

The expression of *grp40L* was not significantly affected by the treatments or sampling points (Figure 4B). However, compared to the control, it was higher before the challenge in the 10 mM SP and 1 mM SB groups.

## 4. Discussion

Intestinal explants allow for the testing of several conditions at the same time and highly reduce the number of experimental animals used in the assays. Thus, ex vivo experiments are a good approach for screening feed additive effects before designing in vivo trials [24].

Despite being in an optimal culture medium with appropriate additives, every explant has a limited lifespan, and the explant viability must be assessed in every experiment [24]. In this study, tissue viability tests showed that independent of the concentration (1 and 10 mM), the SCFAs tested were not toxic to the intestinal explants. However, it was observed that during the challenge, the viability of explants treated with 10 mM was lower than that of explants treated with 1 mM. This suggests that SCFAs tested at higher concentrations might have significantly decreased the medium’s pH and damaged the cells.

Another possible explanation is that the challenging medium was not supplemented with nutrients, and as the intestine is a highly dynamic tissue, this lack of nutrients might have induced cell mortality.

It is well known that the intestine plays a significant role in the host’s immune responses and pathogen elimination via physical barriers and chemical and cellular responses, which ultimately prevent infection [32]. Inflammation is a part of this innate response. After contact with phagocytes, pathogens are engulfed and trapped within an intracellular vesicle, which is then destroyed by digestive enzymes [33]. The production of reactive oxygen species (ROS) is very important for killing pathogens and the evolution of inflammatory reactions, being involved in the activation of several pro-inflammatory cytokines [34]. However, ROS are also responsible for inducing oxidative stress that must be counteracted by the action of antioxidant enzymes and other molecules [35].

In animals, it is known that the immune response and health status can be modulated by environmentally friendly approaches, for example, by using SCFAs as functional feed additives [14]. SCFAs may protect animals by inducing the expression of antioxidant enzymes and immune-related molecules [36]. In fish, information regarding the beneficial effects of SCFA is still limited. However, the available information confirms the beneficial effects of SCFAs. For instance, propionate was shown to promote the immune response and antioxidant defense of zebrafish [3], butyrate was shown to improve the immune response of the common carp [16], and acetate was shown to have beneficial effects on the antioxidant ability and immune function of golden pompano [11].

In the present study, compared to the control, the higher concentrations of propionate and butyrate promoted the expression of *IL-8* expression before the challenge, while propionate also promoted the expression of caspase 3. On the other hand, the lower concentrations of acetate and butyrate promoted the expression *NF-kβ*. An increase in the expression of both inflammatory and anti-inflammatory responses by butyrate was also observed in the crucian carp (*Carassius carassius*) and the Nile tilapia. In the crucian carp, there was an increase in the expression of *IL-8* and *TGF-β* cytokines [37], and in the Nile tilapia, there was an increase in the expression of *TNF-α* and *IL-10* cytokines [38]. In the common carp, acetate slightly increased the expression of the pro-inflammatory cytokines *TNF-α* and *IL-8*, and in goldfish (*Carassius auratus*), a synergistic effect of propionate and butyrate promoted the expression of *TNF-α* and *IL-8* [39].

After the bacterial challenge with *Vibrio anguilarum*, compared to the control, an up-regulation of *TNF-α* was observed with SP 10 mM, while SB 1 mM up-regulated the expression of *IL-8* and *TNF-α*. Moreover, an increase in the expression of the pro-inflammatory cytokines *TNF-α* and *IL-8* was observed in all the experimental groups. Previously, it was also seen that butyrate also promoted the up-regulation of *TNF-α* and *IL-8* in rainbow trout (*Oncorhynchus mykiss*) challenged with *Streptococcus iniae* [32].

In contrast to the responses to the pro-inflammatory cytokines, in the present study, the expression of anti-inflammatory cytokines was either up-regulated (*TGF-β*) or down-regulated (*IL-10*) after the bacterial challenge. Anti-inflammatory cytokines play a crucial role in the control of the immune system by limiting an inflammatory response that may otherwise result in tissue damage, particularly in the gastrointestinal tract [37,40]. Thus, a dynamic balance between pro- and anti-inflammatory mediators must exist and change over time to achieve pathogen control without causing excessive inflammation and tissue damage. This dynamic balance is not equally divided between pro- and anti-inflammatory mediators, and the responses vary with time [41]; this may explain the different responses observed in this study.

SCFAs are known to have antioxidant effects [42]. However, in our study, during the pre-challenge, SCFAs did not affect the expression of the main antioxidant-related enzymes (*cat*, *sod*, and *gpx*). However, in zebrafish, propionate promoted the down-regulation of *sod* and *cat* expression [14], and in yellow catfish (*Pelteobagrus fulvidraco*), a mixture of SCFAs (fumaric, benzoic, and butanoate) decreased *sod* and *cat* activities [43]. However, in contrast to those studies, apple cider vinegar increased *cat* and *gpx* expression in zebrafish [44], and acetate increased the expression of gpx in the common carp [45]. Butyrate was also shown to increase the activity of *sod*, *cat*, and *gpx* in yellow catfish [42].

In the present study, after the bacterial challenge, there was a down-regulation of *cat* and *sod* expression in all the experimental groups (except in the higher concentration of acetate) and an up-regulation of *gpx* expression (except for the lower propionate concentration). In contrast, rainbow trout fed with butyrate-supplemented diets and challenged with *Streptococcus iniae* showed an up-regulation of *sod*, *cat*, and *gpx* expression [32]. Similarly, Nile tilapia fed with butyrate-supplemented diets showed increased *sod*, *cat*, and *gpx* activities after heat stress [19].

In mammals, it was shown that SCFAs can regulate innate immune and inflammatory responses and energy metabolism by binding to specific G-protein coupled receptors (GPCRs), namely GPR41, GPR43, and GPR109A. These three receptors are expressed in the intestinal epithelium and immune cells [20,46,47]. It is also known that, in mammals, the recognition of SCFAs by GPR41 and GPR43 plays an important role in immune function in the intestinal tract [30].

Perit and Wiegertjes [30] identified and validated the coding sequences of *gpr40L* genes, which are very similar to GPR43 in mammals, in 25 different teleost fish species, including the European sea bass. In our study, the expression of *grp40L* was not affected by SCFAs or the bacteria challenge. This is in line with the immune gene expression results, suggesting that SCFAs did not have a significant effect in inducing the immune response in the intestinal explants. However, further studies of the identification and characterization of SCFA receptors and their responses to different SCFAs in fish are needed to better understand their mechanistic action [30].

The SCFA absorbed in the intestine can be directly used by the enterocytes as an energy source or for anabolic processes [48]. It is also known that, in humans, butyrate is a major energy source in the intestine [49]. Citrate synthase (*cs*) is the first enzyme of the Krebs cycle and the route of entry for SCFA catabolism. In our study, *cs* expression before the challenge was not affected by the SCFA source, but compared to the control, it was higher with the higher concentration of propionate, indicating that this SCFA might have been used as a preferential energy source. After the bacterial challenge, *cs* expression was significantly reduced, indicating that the SCFA might have been diverted to immunity modulation instead of being used for energy production.

## 5. Conclusions

Overall, the results of the present study indicate that the SCFA assayed, particularly, butyrate and propionate, affected immune-related gene expression both before and after the bacteria challenge but did not affect the expression of oxidative stress genes. However, further studies are necessary to better understand how SCFA modulates the immune and oxidative responses in fish.

## Figures and Tables

**Figure 1 animals-14-01360-f001:**
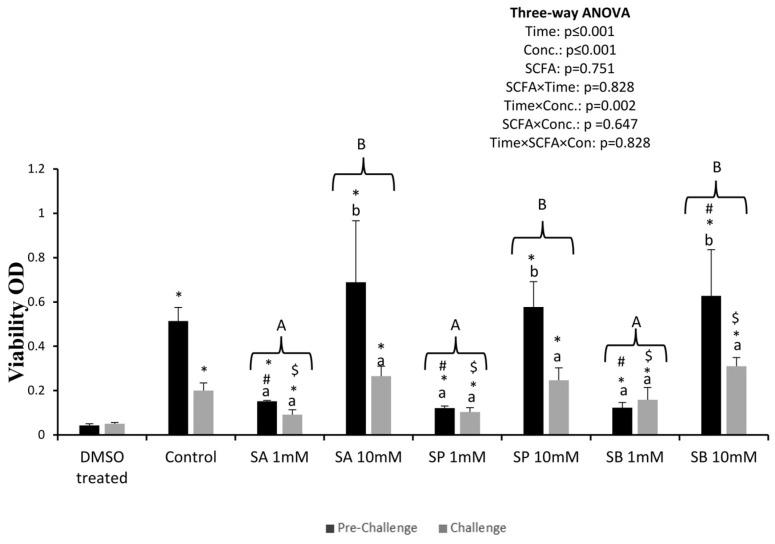
Viability indices (abs/mg) of anterior intestinal explants of the European sea bass. Values are presented as mean (n = 3) and standard error (SE); a three-way ANOVA was performed, followed by a Tukey’s test. Capital letters mean differences between concentrations (1 and 10 mM) within each SCFA; small letters mean differences between the pre-challenge and the challenge within each treatment. Non-orthogonal contrasts were performed each time; # means differences between the control and the SCFA in the pre-challenge, $ means differences between the control and the SCFA in the challenge, and * means differences between explants treated with DMSO and the other treatments.

**Figure 2 animals-14-01360-f002:**
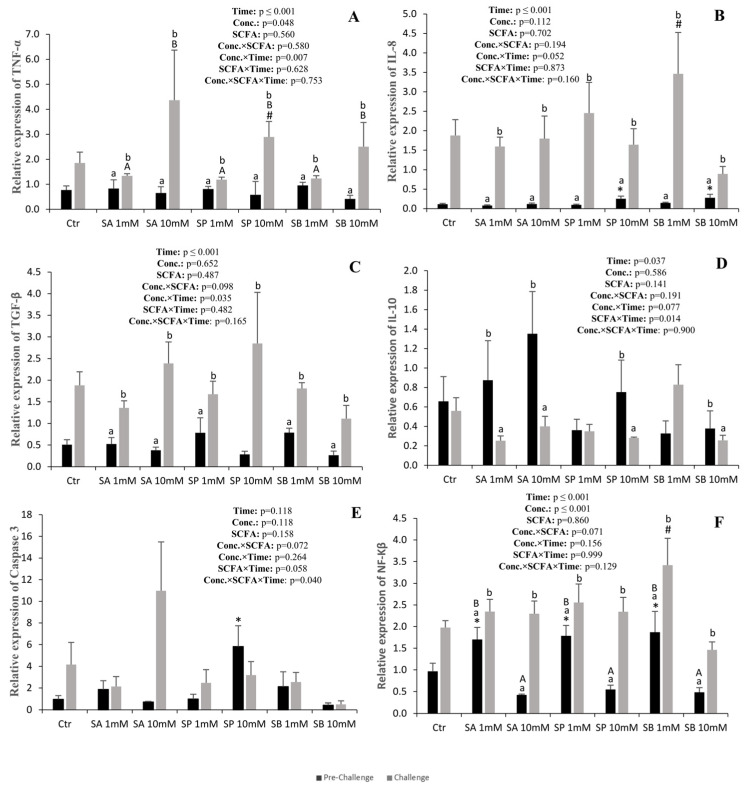
Quantitative expression of TNF-α (**A**), IL-8 (**B**), TGF-β (**C**), IL-10 (**D**), Caspase 3 (**E**) and NF-kβ (**F**) of anterior intestinal explants of European sea bass subjected to the different treatments before (pre-challenge) and after the challenge with *Vibrio anguillarum*. Values are presented as mean (n = 6) and standard error (SE); a three-way ANOVA was performed, followed by a Tukey’s test. Capital letters mean differences between concentrations within each SCFA (1 and 10 mM); small letters mean differences between the pre-challenge and the challenge within each treatment. Non-orthogonal contrasts were made to compare the control with the other treatments: * denotes differences at the pre-challenge, and # denotes differences after the challenge.

**Figure 3 animals-14-01360-f003:**
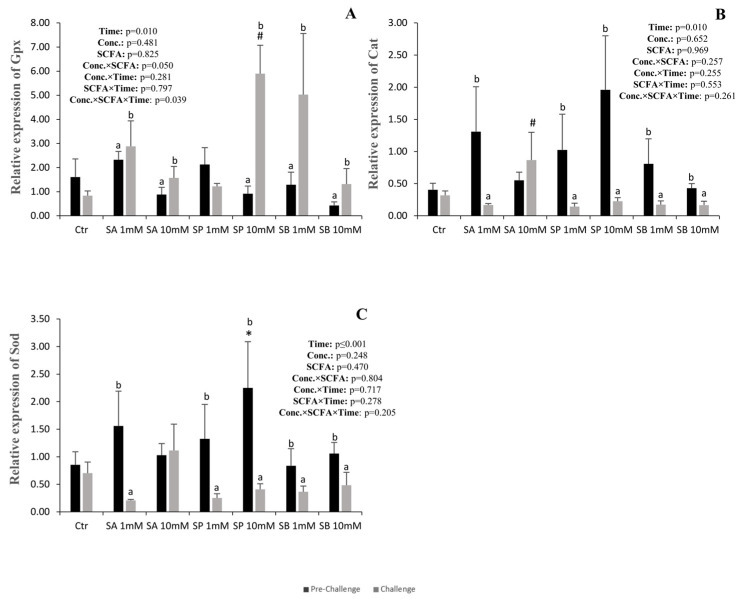
Quantitative expression of Gpx (**A**), Cat (**B**) and Sod (**C**) of anterior intestinal explants of European sea bass subjected to the different treatments, before (pre-challenge) and after challenge with *Vibrio anguillarum*. Values are presented as mean (n = 6) and standard error (SE); a three-way ANOVA was performed, followed by a Tukey’s test. Small letters mean differences between the pre-challenge and the challenge within each treatment. Non-orthogonal contrasts were made to compare the control with the other treatments: * denotes differences at the pre-challenge, and # denotes differences after the challenge.

**Figure 4 animals-14-01360-f004:**
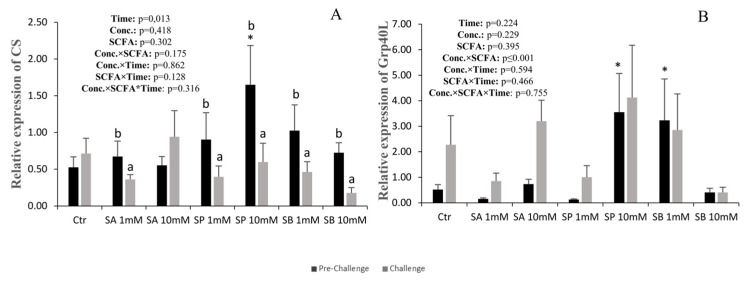
Quantitative expression of energy metabolism-related gene CS (**A**) and the SCFA receptor Grp40L (**B**) of anterior intestinal explants of European sea bass subjected to the different treatments before (pre-challenge) and after the challenge with *Vibrio anguillarum*. Values are presented as mean (n = 6) and standard error (SE); a three-way ANOVA was performed, followed by a Tukey’s test. Small letters mean differences between the pre-challenge and the challenge within each treatment. Non-orthogonal contrasts were made to compare the control with the other treatments: * denotes differences at the pre-challenge.

**Table 1 animals-14-01360-t001:** Primer sequences used for real-time quantitative PCR.

Gene	Gene Abbreviation	Primer Sequences (5′→3′)	Primer Efficiency	Anel. Temperature	Accession Number
Pro-inflammatory					
Tumor necrosis factor-α	*TNFα*	F: AGCCACAGGATCTGGAGCTA R: GTCCGCTTCTGTAGCTGTCC	2.1	60 °C	DQ200910
Interleukin 8	*Il-8*	F: GTCTGAGAAGCCTGGGAGTG R: GCAATGGGAGTTAGCAGGAA	2.0	60 °C	AM490063
Anti-inflammatory					
Transforming growth factor-β	*TGF*-*β*	F: GACCTGGGATGGAAGTGGAT R: CAGCTGCTCCACCTTGTGTTG	2.0	60 °C	AM421619.1
Interleukin 10	*Il-10*	F: ACCCCGTTCGCTTGCCA R: ATCTGGTGACATCACTC	2.0	60 °C	AM268529
Apoptotic					
Caspase 3	*Casp3*	F: CTGATTTGGATCCAGGCATT R: CGGTCGTAGTGTTCCTCCAT	2.1	60 °C	DQ345773
Transcription factor					
Nuclear factor kappa β	*NF-kβ*	F: GCTGCGAGAAGAGAGGAAGA R: GGTGAACTTTAACCGGACGA	1.9	60 °C	DLAgn_00239840 [29]
SCFA receptor					
G-protein coupled receptor	*Grp40L*	F:TTCTGTCCAAACTGCAGCAC R: TCTTACAGCGGAGGAGGAGA	2.0	60 °C	[30]
Oxidative stress					
Catalase	*Cat*	F: ATGGTGTGGGACTTCTGGAG R: AGTGGAACTTGCAGTAGAAACG	1.9	60 °C	FJ860003.1
Superoxide dismutase	*Sod*	F: CATGTTGGAGACCTGGGAGA R: TGAGCATCTTGTCCGTGATGT	2.0	60 °C	FJ860004.1
Glutathione peroxidase	*Gpx*	F:AGTTCGTGCAGTTAATCCGGA R: GCTTAGCTGTCAGGTCGTAAAAC	2.0	60 °C	[31]
Energy metabolism					
Citrate synthase	*Cs*	F: TAGGCAGTGGCATCCAAAGG R: AAGTTGACAGTCTTGATTGGAGC	2.0	60 °C	KF857304.1
Housekeeping					
Elongation factor 1α	*Ef1α*	F: GCTTCGAGGAAATCACCAAG R: CAACCTTCCATCCCTTGAAC	2.0	60 °C	AJ866727
Ribosomal protein S40	*40s*	F: TGATTGTGACAGACCCTCGTG R: CACAGAGCAATGGTGGGGAT	1.9	60 °C	HE978789.1

## Data Availability

The data presented in this study are available on request from the corresponding author.

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
