# Peer review of "The Effects of Short-Chain Fatty Acids in Gut Immune and Oxidative Responses of European Sea Bass (Dicentrarchus labrax): An Ex Vivo Approach"

_animals, 2024, doi:10.3390/ani14091360_

Round 1

Reviewer 1 Report

Comments and Suggestions for Authors

My Decision on the manuscript with ID (animals-2936719) is “Major Revisions”. The authors should prepare a point-by-point response to the comments raised by the anonymous reviewer before the manuscript is considered for publication in Animals.

Throughout the whole manuscript: -

1. You should not repeat the Latin names of fish species after their first appearance in the text.

2. All Latin names should be written in italics.

3. Use abbreviations (gene names, fish, and bacterial species); you should not repeat any name after its first appearance in the text.

4. All gene abbreviations should be written italic.

Missed references.

Line 69: Xun et al. (2022)

Line 74: Wassef et al. (2020)

Line 75: Safari et al. (2016)

Line 80: Liu et al. (2014)

Line 198: Vandesompele et al. (2002).

Discussion:

Line 282- 284: These lines should be supported with appropriate references that are related to the text.

Delete the results from the discussion section:

Line 365: (Figure 4B)

Line 376: (Figure 4A)

Reference section:

1. The authors should follow the MDPI guidelines for writing the references. I advise using EndNote or any reference management program.

2. All Latin names (bacterial species and fish species) should be written in italics.

3. Revise journal name initials as

Line 435: Aquaculture International

Line 438: Fish Physiology and Biochemistry

Specific questions: -

Q1. Line 124: 1x107 CFU/mL. How did you adjust this bacterial dose? Write the methodology. Why did you select this bacterial dose for your experimental work?

Q2. Line 130: Add details on the commercial diet used.

Q3. Line 138: Add the dose of 2-phenoxyethanol that is used for induction on euthanasia. Add appropriate reference for the dose used.

Q4. Table 1.

There are several questions in Table 1.

1. The accession number (DLAgn_00239840) was not found in the NCBI GenBank database. This will result in invalid results.

2. The authors used NCBI GenBank accession numbers of primers designed by other researchers. In your case, you should add references that related to each primer. Please double check the GenBank database, you will find references.

3. For confirming the validity of your results, several data should be presented in this table such as product size (bp), primer efficiency, R2, and Pearson’s coefficient. For replication of the results, you should add Tm and annealing temperature.

Comments on the Quality of English Language

Extensive editing of English language required

Author Response

Thank you very much for taking the time to review this manuscript. Please find the detailed responses below and the corresponding corrections highlighted in the re-submitted files.

Response to Reviewer 1 Comments

Throughout the whole manuscript: -

  1. You should not repeat the Latin names of fish species after their first appearance in the text.
  2. All Latin names should be written in italics.
  3. Use abbreviations (gene names, fish, and bacterial species); you should not repeat any name after its first appearance in the text.
  4. All gene abbreviations should be written italic.

Response: We apologize for the misunderstanding, and it has already been corrected.

Missed references.

Line 69: Xun et al. (2022)

Line 74: Wassef et al. (2020)

Line 75: Safari et al. (2016)

Line 80: Liu et al. (2014)

Line 198: Vandesompele et al. (2002).

Response: We apologize for the misunderstanding, and it has already been corrected.

Discussion:

Line 282- 284: These lines should be supported with appropriate references that are related to the text.

Delete the results from the discussion section:

Line 365: (Figure 4B)

Response: We apologize for the misunderstanding, and it has already been corrected.

Line 376: (Figure 4A)

Response: We apologize for the misunderstanding, and it has already been corrected.

Reference section:

  1. The authors should follow the MDPI guidelines for writing the references. I advise using EndNote or any reference management program.
  2. All Latin names (bacterial species and fish species) should be written in italics.
  3. Revise journal name initials as

Line 435: Aquaculture International

Line 438: Fish Physiology and Biochemistry

Response: We apologize for the misunderstanding, and it has already been corrected.

Specific questions:

Q1. Line 124: 1x107 CFU/mL. How did you adjust this bacterial dose? Write the methodology. Why did you select this bacterial dose for your experimental work?

Response Q1: We thank the reviewer for the comment. The dose was chosen based on previous studies performed in our laboratory. The bacterial dose was adjusted according to the following protocol: The V. anguillarum inoculum concentration was estimated based on the V. anguillarum growth phase, previously determined in the laboratory. Briefly, V. anguillarum was grown in BHI following the same conditions as described in the manuscript: 24h at 28ºC. At different time points of the bacterial growth (every 2h) the optical density (OD600) and the Colony Forming Units (CFU) of the culture were measured to produce a standard curve showing the relationship between the CFUs and OD. This curve is then used to quickly estimate the number of cells presented in the V. anguillarum culture, and consequent adjustment to the desired concentration. Additionally, the number of viable cells used in the final assay is always confirmed by plating serial dilutions of the bacterial suspension in BHI.

Q2. Line 130: Add details on the commercial diet used.

Response Q2: We thank the reviewer for the comment. Commercial diets details was added (Aquasoja Sustainable Feed-Standart Orange 4 MEO4; Sorgal, Ovar, Portugal; analytical constituents: 44% crude protein, 18% crude fat 2.5% fiber, 11.1% ash) in lines 130-132.

Q3. Line 138: Add the dose of 2-phenoxyethanol that is used for induction on euthanasia. Add appropriate reference for the dose used.

Response Q3: We thank the reviewer for the comment. The dose used for euthanasia induction was 1ml/L, according to Martins, N., et al., Dietary oleic acid supplementation improves feed efficiency and modulates fatty acid profile and cell signaling pathway in European sea bass (Dicentrarchus labrax) juveniles fed high-lipid diets. Aquaculture, 2023. 576: p. 739870.

Q4. Table 1.

There are several questions in Table 1.

1.The accession number (DLAgn_00239840) was not found in the NCBI GenBank database. This will result in invalid results.

  1. The authors used NCBI GenBank accession numbers of primers designed by other researchers. In your case, you should add references that related to each primer. Please double check the GenBank database, you will find references.

Response 1 and 2: We thank the reviewer for the comment.

The primers were designed in Primer3  and the quality was analyzed with the Beacon Designer Program. Primer efficiency was validated with serial two-fold dilutions of cDNA and calculated from the slope of the regression line of the quantification cycle (Ct) versus the relative concentration of cDNA. NF-kβ was found in European sea bass genome database (http://seabass.mpipz.mpg.de/cgi-bin/hgGateway). The information was added in line 198-202

  1. For confirming the validity of your results, several data should be presented in this table such as product size (bp), primer efficiency, R2, and Pearson’s coefficient. For replication of the results, you should add Tm and annealing temperature.

Response 3: We thank the reviewer for the comment. Relevant data have been added to the table 1.

Reviewer 2 Report

Comments and Suggestions for Authors

The study evaluating the intestinal interactions between Short-Chain Fatty Acids (SCFA) and pathogenic bacteria in European sea bass juveniles presents intriguing findings that underscore the potential of SCFAs, particularly sodium propionate and sodium butyrate, in enhancing the immune response at the intestinal level. This research sheds light on the intricate dynamics within the fish intestine and offers valuable insights into potential strategies for bolstering immune defense mechanisms. The topic discussed offers valuable insights and perspectives. With some minor adjustments, it could be an impactful addition to the publication.

Specific line comments:

Lines 11-18: I suggest improving the simple summary; currently, it results in sentences detached from each other.

L27- 124: Change the number “7” in superscript.

Line 86: In my opinion, the introduction is well-written but is a little bit too long.  

Line 130 and 132: You have double spaces in sentences.

Table 1: I suggest removing the first column, you have indicated the complete name previously in the text (abstract). Exception for a few of them can be insert in table caption.

Results: you have explained the results, but the graphs are too small, particularly figure 2.

The legends in all figures are not immediately visible, too small, and not well-defined.

Figure 1, the y-axes should be indicate “viability [OD]”

I personally found the discussion very long.

Line 290: The study suggests that higher concentrations of short-chain fatty acids (SCFA) might decrease explant viability due to a decrease in medium pH and potential cell damage. However, it would be essential to delve deeper into the mechanisms underlying this observation. Was the pH of the challenging medium monitored throughout the experiment?

References: You should modify the references. There are duplicate numbers.

Author Response

Thank you very much for taking the time to review this manuscript. Please find the detailed responses below and the corresponding corrections highlighted in the re-submitted files.

Response to Reviewer 2 Comments

Specific line comments:

Lines 11-18: I suggest improving the simple summary; currently, it results in sentences detached from each other.

L27- 124: Change the number “7” in superscript.

Response: We thank the reviewer for the comment. It has already been corrected.

Line 86: In my opinion, the introduction is well-written but is a little bit too long.  

Response: We thank the reviewer for the comment. After submitting an initial version, the editor requested that the word count be increased to 4000, according to journal guidelines.

Line 130 and 132: You have double spaces in sentences.

Response: We thank the reviewer for the comment. It has already been corrected.

Table 1: I suggest removing the first column, you have indicated the complete name previously in the text (abstract). Exception for a few of them can be insert in table caption.

Response: We thank the reviewer for the comment. The first column of table 1 was a well considered option. The authors consider that this helps to guide the readers.

Results: you have explained the results, but the graphs are too small, particularly figure 2.

Response: We thank the reviewer for the pertinent. We have increased the size of figure 1.

The legends in all figures are not immediately visible, too small, and not well-defined.

Response: We thank the reviewer for the comment. We have increased the size of the legends.

Figure 1, the y-axes should be indicate “viability [OD]”

I personally found the discussion very long.

Response: We thank the reviewer for the comment. After submitting an initial version, the editor requested that the word count be increased to 4000, according to journal guidelines.

Line 290: The study suggests that higher concentrations of short-chain fatty acids (SCFA) might decrease explant viability due to a decrease in medium pH and potential cell damage. However, it would be essential to delve deeper into the mechanisms underlying this observation. Was the pH of the challenging medium monitored throughout the experiment?

We thank the reviewer for the comment. Unfortunately, we did not monitored the pH during the experiment, and we provide just a tentative explanation based on the acidic nature of the SCFA.

References: You should modify the references. There are duplicate numbers.

Reviewer 3 Report

Comments and Suggestions for Authors

The result and the discussion need to be improved. Please find the attachment for more details.

Comments on the Quality of English Language

Some minor grammar issues.

Author Response

Thank you very much for taking the time to review this manuscript. Please find the detailed responses below and the corresponding corrections highlighted in the re-submitted files.

Response to Reviewer 3 Comments

Abstract

  1. L27: What’s the unit of 1×107? 1×107 or 1×107?

Response: We thank the reviewer for the comment. It has already been corrected.

Introduction

  1. All scientific names should be italics

Response: We thank the reviewer for the comment. It has already been corrected.

  1. L46-48: Please provide citation. Currently, antibiotics are not often used, so please provide reference or any data to support the statements.

Response: We thank the reviewer for the comment. It has already been corrected.

  1. L54: “acteristics. the most abundant SCFA…” Should be “ T ”

Response: We thank the reviewer for the comment. It has already been corrected.

  1. L72-73: “the addition of (100 and 72 200mmol/L) of sodium acetate to the diets…” Only

need one “of”

Response: We thank the reviewer for the comment. It has already been corrected.

  1. L78: “Furthermore, Juvenile largemouth bass…” Should be “ j ”

Response: We thank the reviewer for the comment. It has already been corrected.

  1. L101-105: Please combine them together as one paragraph.

Response: We thank the reviewer for the comment. It has already been corrected.

  1. L107-114: Please combine them together as one paragraph.

Materials and methods

  1. L124: 1x107 cfu/ml or 1x107

cfu/ml ?

Response: We thank the reviewer for the comment. It has already been corrected.

  1. L152: vibrio concentration unit?

Response: We thank the reviewer for the comment. Colony Forming Units (CFUs) mL-1 was added to the text line 154.

  1. L152-154: Please combine into L144-151. Do not use a single sentence as a

paragraph.

Response: We thank the reviewer for the comment. It has already been corrected.

  1. L183: double check “H2O” or “H2O”

Response: We thank the reviewer for the comment. It has already been corrected.

Results

  1. All sections in Results need improvement in expressing the results and the figure.

They are too simplistic in the in the current version. The author used three-way

ANOVA, yet I did not find any description of the statistical analysis.

Response: We thank the reviewer comment. However, the simplistic way of describing the results was a well considered option. Figures are self-explanatory but we recognize that complex to fully understand.  The 3-way ANOVA data is also summarized in the figures. To describe in detail all results would be fastidiously and of not much use to the reader. Therefore, in the text, we try to summarize results and guide the readers through the complex amount of data presented in the figures by just highlighting the significant differences observed.

Discussion

  1. L340-355: The author mentioned the result trends in their study as well as those

from previous research, which is good. However, they did not further explain the

potential reasons for the result trends or why there are differences between their

results and those of other researchers

Response: Our results suggest that, contrary to what was observed in other studies,  SCFA did not affect the oxidative stress response, neither before or after the challenge. Oxidative stress response is a complex system that involves enzymes and other molecules and response is not necessarily similar in all organisms or in different tissues within an organism. Furthermore, our study is ex vivo, so the response may be different. More studies are needed to fully understand the effect of SCFA in the antioxidative response of European sea bass.

Round 2

Reviewer 1 Report

Comments and Suggestions for Authors

I advise against the publication of the manuscript with ID (animals-2936719-peer-review-v2). The authors did not appropriately respond to questions raised by the anonymous reviewer. The accession number (DLAgn_00239840) is not found in the NCBI GenBank. The accession number (FM013606.1) is not a correct one for Glutathione peroxidase gene of European seabass. Besides, the manuscript needs extensive English Editing and proof reading. Reference section needs extensive revisions as I illustrated previously in R1.

Comments on the Quality of English Language

Extensive editing of English language required

Author Response

Comments reviewer 1 round 2

I advise against the publication of the manuscript with ID (animals-2936719-peer-review-v2). The authors did not appropriately respond to questions raised by the anonymous reviewer. The accession number (DLAgn_00239840) is not found in the NCBI GenBank. The accession number (FM013606.1) is not a correct one for Glutathione peroxidase gene of European seabass. Besides, the manuscript needs extensive English Editing and proof reading. Reference section needs extensive revisions as I illustrated previously in R1.

Thank you for the comments. We revised the English of the manuscript and we think that it now meets the scientific English standards. We apologize for some minor errors in the reference section. We now thoroughly reviewed it and hope that it meets the Journal requirements.

We do not agree with your advise for not publishing the paper based in two minor disagreements regarding two gene primers.

As we said in the previous comments to reviewers, the gene DLAgn_00239840 is not in the NCBI GeneBank but it can be found in European sea bass genome database (http://seabass.mpipz.mpg.de/cgi-bin/hgGateway). This database is publicly available in the link http://seabass.mpipz.mpg.de  and this database has already been used in other publications (for instance,[1,2]).

We agree that the accession number (FM013606.1) is not correctly described in GeneBank for Glutathione peroxidase gene of European seabass. However, the primer we used is the one presented in papers [3,4] and that indicate this GeneBank entry.

To avoid confusion, in this revised version we indicate as reference the paper already published in Animals.

  1. Ferreira, I.A.; Peixoto, D.; Losada, A.P.; Quiroga, M.I.; Vale, A.d.; Costas, B. Early innate immune responses in European sea bass (Dicentrarchus labrax L.) following Tenacibaculum maritimum infection. Frontiers in Immunology 2023, 14, doi:10.3389/fimmu.2023.1254677.
  2. Toubanaki, D.K.; Efstathiou, A.; Tzortzatos, O.P.; Valsamidis, M.A.; Papaharisis, L.; Bakopoulos, V.; Karagouni, E. Nervous Necrosis Virus Modulation of European Sea Bass (Dicentrarchus labrax, L.) Immune Genes and Transcriptome towards Establishment of Virus Carrier State. International journal of molecular sciences 2023, 24, doi:10.3390/ijms242316613.
  3. Torrecillas, S.; Terova, G.; Makol, A.; Serradell, A.; Valdenegro-Vega, V.; Izquierdo, M.; Acosta, F.; Montero, D. Dietary Phytogenics and Galactomannan Oligosaccharides in Low Fish Meal and Fish Oil-Based Diets for European Sea Bass (Dicentrarchus labrax) Juveniles: Effects on Gill Structure and Health and Implications on Oxidative Stress Status. Frontiers in Immunology 2021, 12, doi:10.3389/fimmu.2021.663106.
  4. Ceccotti, C.; Al-Sulaivany, B.S.A.; Al-Habbib, O.A.M.; Saroglia, M.; Rimoldi, S.; Terova, G. Protective Effect of Dietary Taurine from ROS Production in European Seabass under Conditions of Forced Swimming. Animals 2019, 9, 607.

Reviewer 3 Report

Comments and Suggestions for Authors

Author Response

Comments reviewer 3 round 2

The manuscript has been improved but should be further improved to achieve the quality of acceptance.

 Abstract 1. L27: What is the unit of 1×107 ? cfu/ ml or what?

Response: We apologize for the misunderstanding, and it has already been corrected.

Materials and methods 1.

L130: scientific name should be italics --- Please double check through out the manuscript 2.

Response: We apologize for the misunderstanding, and it has already been corrected.

 L154: Please uniformed the unit expression used in your manuscript. You already used cfu/ml before in L124.

Response: We apologize for the misunderstanding, and it has already been corrected.
